# New Meroterpenes from South China Sea Soft Coral *Litophyton brassicum*

**DOI:** 10.3390/md22090392

**Published:** 2024-08-30

**Authors:** Xiaoyi Chen, Jiahui Zhang, Jiayu Yang, Bing Li, Te Li, Han Ouyang, Wenhan Lin, Hongyu Hu, Xia Yan, Shan He

**Affiliations:** 1Li Dak Sum Yip Yio Chin Kenneth Li Marine Biopharmaceutical Research Center, Ningbo University, Ningbo 315211, China; chennimo@163.com (X.C.); 13735210075@163.com (J.Z.); yangjiayv1210@163.com (J.Y.); 18868105112@163.com (B.L.); telinbu@163.com (T.L.); 2Institute of Drug Discovery Technology, Ningbo University, Ningbo 315211, China; ouyanghan@nbu.edu.cn; 3Ningbo Institute of Marine Medicine, Peking University, Ningbo 315800, China; whlin@bjmu.edu.cn; 4Xingzhi College, Zhejiang Normal University, Lanxi 321004, China; huhongyu22@126.com

**Keywords:** soft coral, *Litophyton brassicum*, meroterpenes, antibacterial activity

## Abstract

A chemical investigation of the extracts from the soft coral *Litophyton brassicum* led to the isolation and identification of four new meroterpenes, brassihydroxybenzoquinone A and B (**1** and **2**) and brassinaphthoquinone A and B (**3** and **4**), along with two known related meroterpenes (**5** and **6**). Their structures were elucidated using high-resolution electrospray ionization mass spectrometry (HRESIMS), nuclear magnetic resonance (NMR) spectroscopy, and a comparison with the literature data. All compounds were evaluated for antibacterial activity against six pathogenic bacterial strains and for cytotoxic activity against three cancer cell lines. In the cytotoxic assay, all compounds were inactive at 10 μM against the A549, HeLa, and MDA-MB-231 cell lines. In the antibacterial assay, compounds **1** and **2** exhibited moderate inhibitory activity with minimum inhibitory concentrations (MIC) ranging from 8 to 64 μg/mL.

## 1. Introduction

The marine soft coral genus *Litophyton* belongs to the family Nephtheidae, order Alcyonacea, subclass Octocorallia. Currently, *Litophyton* comprises nearly 100 species, widely distributed throughout tropical and temperate waters such as the South China Sea, the Red Sea, and other regions of the Indo-Pacific Ocean [1]. Chemical investigations on *Litophyton* soft corals have revealed them to be prolific producers of bioactive secondary metabolites. Since the early 1970s, when novel cembrane diterpenes were first reported from *L. viridis* [2], numerous research groups worldwide have conducted chemical investigations on *Litophyton*, leading to significant discoveries. To date, nearly 200 secondary metabolites have been isolated and characterized from *Litophyton* corals over 50 years of research [1,3]. These compounds include sesquiterpenes, sesquiterpene dimers, diterpenes, norditerpenes, tetraterpenes, meroterpenes, steroids, ceramides, pyrimidines, and peptides [1]. A broad spectrum of pharmacological activities has been evaluated, including cytotoxic [4,5,6,7], antiviral [5], antibacterial [8], antifungal [9], antimalarial [10], anti-inflammatory [11], PTP1B inhibitory [12,13].

Meroterpenes, a class of compounds found in *Litophyton*, are relatively rare in the literature. For example, four meroterpenes isolated from the Red Sea soft coral *Nephthea* sp. were identified as potential SARS-CoV-2 main protease inhibitors [14]. Nine new meroditerpenoid-related metabolites were isolated from the Formosan soft coral *Nephthea chabrolii*, with proposed biosynthetic pathways [15]. Then, the next year, eight new meroditerpenoid-related metabolites were isolated from the organic extract of a Taiwanese soft coral *Nephthea chabrolii* [16]. It is noteworthy that the genus *Nephthea* was synonymized with *Litophyton* in 2016 due to their identical characteristics [1]. As part of our ongoing efforts to discover bioactive marine natural products from soft corals, specimens of *Litophyton brassicum* were collected from the South China Sea. Herein, we report the isolation and identification of six meroterpenes. The antibacterial activities of these compounds were evaluated against six pathogenic bacteria, and their antiproliferative effects were tested on three cancer cell lines.

## 2. Results

The acetone extract of soft coral *Litophyton brassicum* was subjected to repeated silica gel and reversed-phase silica gel column chromatography, followed by semipreparative HPLC, to afford four new meroterpenes (**1**–**4**) and two known compounds (**5** and **6**) (Figure 1). The known compounds were identified as chabrolohydroxybenzoquinone G [16] and chabrolohydroxybenzoquinone B [15].

Compound **1** was obtained as an optically active colorless oil. Its molecular formula was determined to be C_26_H_38_O_4_ based on HRESIMS analysis [M-H_2_O + H]^+^ *m*/*z* 397.2755, calcd. for C_26_H_37_O_3_, 397.2742), indicating eight degrees of unsaturation. In the ^1^H NMR spectrum, two aromatic signals resonating at *δ*_H_ 6.55 (1H, br. s) and 6.42 (1H, br. s) indicated the presence of a 1, 2, 4, 5-tetrasubstituted benzene system. Four olefinic protons *δ*_H_ 6.25 (1H, d, *J* = 9.8 Hz), 5.52 (1H, d, *J* = 9.8 Hz), 5.09 (1H, tq, *J* = 7.2, 1.4 Hz), and 5.05 (1H, tq, *J* = 7.4, 1.4 Hz), indicated the presence of regular prenyl moieties. The ^1^H, ^13^C NMR, DEPT, and HSQC spectra data confirmed the presence of 26 carbons, including a ketone carbonyl at *δ*_C_ 211.4, a benzene system, one disubstituted double bond [δ_C_ 122.6 (C-1), δ_C_ 129.9 (C-2)], two trisubstituted double bonds [*δ*_C_ 125.0 (C-6), *δ*_C_ 134.7 (C-7), *δ*_C_ 123.0 (C-14), and *δ*_C_ 132.8 (C-15)], an oxygenated quaternary carbon (*δ*_C_ 78.1), seven methylenes, and five methyls [δ_C_ 2.18 (C-7′), δ_H_ 1.60 (C-16), δ_H_ 1.66 (C-17), δ_H_ 1.55 (C-18), and δ_H_ 1.35 (C-19)]. The constitution of the side chain was elucidated initially by the ^1^H-^1^H COSY correlations of H-1/H-2, H_2_-4/H_2_-5/H-6, H_2_-8/H_2_-9/H_2_-10, and H_2_-12/H_2_-13/H-14 (Figure 2). The spectroscopic data were similar to those of the known compound chabrolohydroxybenzoquinones E [16], except for the missing of a methyl group on C-11, and the Δ^10,11^ double bond was reduced and oxidized to a keto group in **1**. In addition, the HMBC correlations from H_2_-9, H_2_-10, H_2_-12, and H_2_-13 to C-11 implied the keto group was on C-11. The smaller coupling constant (*J*_1,2_ = 9.8 Hz) and the 1D NOE correlation (Figure 3) from H-6 to H_2_-8 suggested the *Z* configuration of the Δ^1,2^ and *E* configuration of the Δ^6,7^ double bonds, respectively. Comprehensive HMBC analysis allowed the complete assignment of the proton and carbon signals for **1** (Table 1 and Figure 2). As a result, the structure of **1** was elucidated as shown in Figure 1, named brassihydroxybenzoquinone A.

Compound **2** (brassihydroxybenzoquinone B) was also obtained as a colorless oil. The molecular formula of C_27_H_40_O_4_ derived by the HR-ESI-MS ion peak of [M-2H_2_O + H]^+^ at *m*/*z* 393.2798 (calcd. for C_27_H_37_O_2_, 393.2793) gave 8 degrees of unsaturation. The ^1^H and ^13^C NMR spectra of **2** (Table 1) were also similar to those of chabrolohydroxybenzoquinones E [16], except that the oxygen-bearing methylene [*δ*_C_ 60.5 (C-18)/*δ*_H_ 4.10] attached at C-11 in **2** rather than the methyl group in chabrolohydroxybenzoquinones E, which was proven by the HMBC correlations from H_2_-18 to C-10 (*δ*_C_ 128.8), C-11 (*δ*_C_ 138.4), and C-12 (*δ*_C_ 35.3). The configuration of the Δ^1,2^ double bond was also assigned as *Z*, based on the same method as for **1**. The *E* geometry of Δ^6,7^ and *Z* geometry of Δ^10,11^ double bonds in **2** were supported by the 1D NOE enhancements of H-6/H_2_-8 and H-10/H_2_-12, respectively. We attempted to utilize the TDDFT-ECD calculation method to establish the absolute configuration of the C-3 position in compounds **1** and **2**. However, the experimental ECD spectrum did not exhibit a clear Cotton effect (Appendix A), which is likely due to the C-3 position being located on the side chain, leading to its flexibility. As far as we know, determining the absolute configuration of C-3 in compounds **1** and **2** remains a challenging task.

Compound **3** (brassinaphthoquinone A) was a yellow oil. The molecular formula of C_27_H_34_O_3_, giving 11 degrees of unsaturation, was established by the HR-ESI-MS ion peak at *m*/*z* 389.2484 [M-H_2_O + H]^+^ (calcd. for C_27_H_33_O_2_, 389.2480). From the ^1^H NMR spectrum of **3**, the resonances of three aromatic protons *δ*_H_ 7.96 (1H, d, *J* = 9.8 Hz), 7.90 (1H, d, *J* = 1.8 Hz), and 7.52 (1H, dd, *J* = 9.8, 1.8 Hz) indicated the presence of a 1, 2, 4-trisubstituted benzene system. One additional aromatic proton 6.81 (1H, br.s) was also observed. From the ^1^H and ^13^C NMR spectral data (Table 1), together with the HSQC data, 27 signals were assigned to two carbonyls, seven sp^2^ quaternary olefinic carbons, seven sp^2^ methine, an oxygenated sp^3^ methine [*δ*_C_ 66.1 (C-9)/*δ*_H_ 4.42], five sp^3^ methylene, and five methyls. The NMR data of **3** closely resembled those of chabrolonaphthoquinone A [15], a meroditerpenoid with a naphthoquinone moiety obtained from the Formosan soft coral *Nephthea chabrolii*. The differences were one more methyl group (C-18) on **3** rather than the carboxyl group on chabrolonaphthoquinone A, and C-9 was hydroxylated on **3**. The different side chain was elucidated by the ^1^H-^1^H COSY correlations of H_2_-8/H-9/H-10 and the HMBC correlations from H-9 to C-7, C-10, and C-11. The key 1D NOE correlations (Figure 3) from H-6 to H_2_-8 and from H-10 to H_2_-12 established the *E* configuration of the Δ^6,7^ and Δ^10,11^ double bonds, respectively. Thus, the structure of **3** was elucidated, as shown in Figure 2.

Brassinaphthoquinone B (**4**) was also isolated as a yellow oil with a molecular formula of C_27_H_34_O_3_ on the basis of HR-ESI-MS ion peak at *m*/*z* 389.2471 [M-H_2_O + H]^+^ (calcd. for C_27_H_33_O_2_, 389.2480), which is the structural isomer of **3**. The ^1^H and ^13^C NMR data (Table 1) of **4** were extremely similar to those of **3**, except that the absence of hydroxyl group at C-9 in **4**, and the oxygenated sp^3^ carbon [*δ*_C_ 60.5 (C-18)/*δ*_H_ 4.11] was a methylene in **4** rather than a methine in **3**. It was deduced by analysis of the ^1^H-^1^H COSY correlations of H_2_-8/H_2_-9/H-10 and the HMBC correlations from H_2_-18 to C-10, C-11, and C-12. The 1D NOE enhancements between H-6/H_2_-8 and H-10/H_2_-12 indicated the *E* geometry of the Δ^6,7^ and Δ^10,11^ double bonds, respectively.

Compounds **1**–**6** were evaluated for cytotoxicity and antibacterial activities. In the cytotoxic assay, all compounds were inactive at 10 *μ*M against the cell lines of A549, HeLa, and MDA-MB-231. An evaluation of the antibacterial activity against six pathogenic bacterial strains (*Staphylococcus aureus*, *Escherichia coli*, *Pseudomonas aeruginosa, Bacillus subtilis*, *Vibrio parahaemolyticus*, and *Vibrio harveyi*) showed that compounds **1** and **2** displayed moderate inhibitory activity (MIC 8–64 μg/mL) (Table 2). Interestingly, compound **2** may have potential as an antibiotic agent for controlling aquatic pathogens in the future.

## 3. Materials and Methods

### 3.1. General Chemical Experimental Procedures

NMR spectra were recorded on a Bruker AVANCE NEO 600 spectrometer (BrukerBiospin AG, Fällanden, Germany). ^1^H chemical shifts were referenced to the residual CDCl_3_ (7.26 ppm), and ^13^C chemical shifts were referenced to the CDCl_3_ (77.2 ppm) solvent peaks. High-resolution electrospray ionization mass spectra (HRESIMS) were performed on an ultra-high-performance liquid chromatograph (UPLC) and TIMS-QTOF high-resolution mass spectrometry (Waters, MA, USA). The purification was performed by reversed-phase high-performance liquid chromatography using a Shimadzu LC-20AT system (Shimadzu Corporation, Tokyo, Japan). The solvents used for HPLC were all Fisher HPLC grade. A Cosmosil C_18_-MS-II column (250 mm × 20.0 mm, id, 5 μm, Cosmosil, Nakalai Tesque Co., Ltd., Kyoto, Japan) was used for the preparative HPLC separation. Column chromatography was performed using silica gel (300–400 mesh, Qingdao Ocean Chemical Co., Ltd., Qingdao, China) and C_18_ reversed-phase silica gel (75 µm, Nakalai Tesque Co., Ltd., Kyoto, Japan).

### 3.2. Animal Material

Soft coral *Litophyton brassicum* was sampled off the coast of Xisha Islands, South China Sea, 12 m underwater, with a wet weight of 5.07 kg, and it was frozen immediately after collection. The specimens (XSSC201906) were deposited at the Li Dak Sum Yip Yio Chin Kenneth Li Marine Biopharmaceutical Research Center, Health Science Center, Ningbo University, China.

### 3.3. Extraction and Isolation

The soft coral samples were vacuum freeze-dried with a freeze-dryer, crushed in a pulverizer, and fully soaked in acetone at room temperature for 2 days each time, followed by ultrasonic extraction for 1 h, repeated soaking ultrasonic extraction for 4–5 times. The extract was filtered to remove the sample residue, and the extract was concentrated under reduced pressure. The extract was partitioned three times with Et_2_O and water (1:1, v:v), and the Et_2_O layer extract was concentrated under reduced pressure to obtain 80 g brown residue.

The 80 g of the extract was separated by gradient elution on a normal-phase silica gel column, yielding 11 fractions (FrA–FrK). Fr.G (1.8 g) was eluted with MeOH/H_2_O (75:15 to 100:0, *v*/*v*) on reversed-phase column chromatography to obtain five subfractions (Fr.G.1–Fr.G.5). Purification of Fr.G.3 by semi-preparative HPLC (MeCN/H_2_O, 72:18, 2 mL/min) gave compounds **1** (6.8 mg, t*_R_* = 62 min) and **3** (5.8 mg, t*_R_* = 71 min). Separation of Fr.H (1.54 g) on a reversed-phase column with MeOH/H_2_O (75:15~100:0, *v*/*v*) afforded seven subfractions (Fr.H.1~Fr.H.6). Fr.H.5 was purified by semipreparative reversed-phase HPLC (MeCN/H_2_O, 75: 15, 2 mL/min) to provide compounds **2** (5.7 mg, t*_R_* = 60 min) and **4** (7.8 mg, t*_R_* = 40 min). Separation of Fr.J (1.4334 g) on a reversed-phase column with MeOH/H_2_O (80:20~100:0, *v*/*v*) provided seven subfractions (Fr.J.1~Fr.J.6). Fr.J.6 was purified by semipreparative reversed-phase HPLC (MeCN/H_2_O, 80:20, 2 mL/min) to provide compounds **5** (6.7 mg, t*_R_* = 53 min) and **6** (8.8 mg, t*_R_* = 62 min).

Brassihydroxybenzoquinone A (**1**): colorless oil; {[α${]}_{D}^{25}$ −17.40 (c 0.5, MeOH)}; UV (MeOH): 221 (3.38), 270 (2.51), 330 (2.57); IR (KBr) *ν* = 3400, 3100, 1766, 1651, 1261, 1079, 888, 630 cm^−1^; ^1^H and ^13^C NMR data, Table 1; HRESIMS *m*/*z* 397.2755 [M-H_2_O + H]^+^ (calcd. for C_26_H_37_O_3_, 397.2742).

Brassihydroxybenzoquinone B (**2**): colorless oil; {[α${]}_{D}^{25}$ −18.67 (c 0.5, MeOH)}; UV (MeOH): 220 (3.27), 270 (2.48), 330 (2.55); IR (KBr) *ν* = 3574, 3154, 1436, 1261, 1037, 951, 710 cm^−1^; ^1^H and ^13^C NMR data, Table 1; HRESIMS *m*/*z* 393.2798 [M-2H_2_O + H]^+^ (calcd. for C_27_H_37_O_2_, 393.2793).

Brassinaphthoquinone A (**3**): yellow oil; {[α${]}_{D}^{25}$ −23.47 (c 0.5, MeOH)}; UV (MeOH): 204 (3.39), 237 (3.23), 270 (3.05); IR (KBr) *ν* = 3582, 3088, 1765, 1692, 1450, 1139, 916, 749 cm^−1^; ^1^H and ^13^C NMR data, Table 1; HRESIMS *m*/*z* 389.2484 [M-H_2_O + H]^+^ (calcd. for C_27_H_33_O_2_, 389.2480).

Brassinaphthoquinone B (**4**): yellow oil; {[α${]}_{D}^{25}$ −23.73 (c 0.5, MeOH)}; UV (MeOH): 204 (3.34), 245 (3.13), 270 (2.90); IR (KBr) *ν* = 3583, 2928, 1759, 1692, 1470, 1138, 915, 749 cm^−1^; ^1^H and ^13^C NMR data, Table 1; HRESIMS *m*/*z* 389.2480 [M-H_2_O + H]^+^ (calcd. for C_27_H_33_O_2_, 389.2481).

### 3.4. Antibacterial Assays

All isolated compounds were tested for antibacterial activities according to established methods [17]. Six bacterial strains were selected: *S. aureus* [CMCC (B) 26003], *B. subtilis* [CMCC (B) 63501], *V. harveyi* 1708B04 (accession number: MZ333451), *V. parahaemolyticus* (accession number: OL636376), *P. aeruginosa* [CMCC (B) 10104], and *E. coli* [CMCC (B) 44102], with penicillin G serving as the positive control. Compounds **1**–**6** were dissolved in DMSO and tested at concentrations of 128, 64, 32, 16, 8, 4, and 2 µg/mL. Briefly, bacteria were grown in Mueller–Hinton (MH) medium for 24 h at 28 °C with agitation (180 rpm), then diluted with sterile MH medium to match the 0.5 McFarland standard. One hundred microliters of each bacterial suspension and 100 μL of MH medium containing 0.002% 2,3,5-triphenyltetrazolium chloride, along with test or control compounds, were incubated. Inhibition data were recorded optically.

### 3.5. Cytotoxic Activity Assays

MDA-MB-231, HeLa, and A549 cells were cultured in DMEM (Gibco, Thermo Fisher Scientific, Inc., Waltham, MA, USA) supplemented with 10% FBS (Gibco, Thermo Fisher Scientific, Inc.). These cell lines were obtained from the Shanghai Cell Bank, Chinese Academy of Sciences. The cells were incubated at 37 °C in a humidified atmosphere containing 5% CO_2_. MTT assays were performed as described by Zhang et al. (2011) [18]. Briefly, cells were seeded into 96-well plates at a density of 5 × 10^3^ cells/well and incubated for 12 h, followed by exposure to various test compounds at different concentrations for 48 h. Subsequently, the cells were stained with 20 μL of MTT solution (5 mg/mL) for 4 h. The medium and MTT solution were then removed, and 150 μL of DMSO was added to dissolve the formazan crystals. The plates were shaken at low speed for 10 min. Absorbance was measured at 490 nm using a microplate reader.

## 4. Conclusions

In summary, the chemical study of the soft coral *Litophyton brassicum* from the South China Sea led to the identification of four novel meroterpenes, brassihydroxybenzoquinone A and B and brassinaphthoquinone A and B, along with two known related meroterpenes. The structures of these compounds were elucidated using HRESIMS and NMR spectroscopy and corroborated with the existing literature. Compounds **1** and **2** exhibited moderate antibacterial activity, with minimum inhibitory concentrations (MIC) ranging from 8 to 64 μg/mL. The isolation of these novel terpenoids from marine soft corals underscores the rich chemical diversity of the *Litophyton genus*. These newly characterized compounds, with their unique structures, hold promise for developing innovative antimicrobial therapies, particularly amid escalating antibiotic resistance. Exploring their antimicrobial properties is, therefore, a critical endeavor for public health.

## Figures and Tables

**Figure 1 marinedrugs-22-00392-f001:**
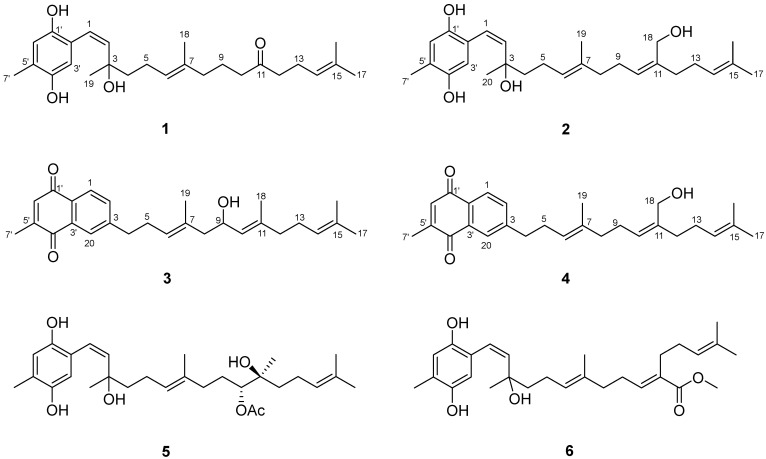
Chemical structures of compounds **1**–**6**.

**Figure 2 marinedrugs-22-00392-f002:**
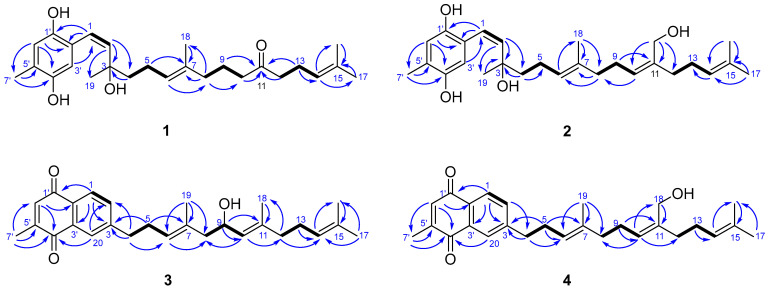
^1^H-^1^H COSY and key HMBC correlations of compounds **1**–**4**.

**Figure 3 marinedrugs-22-00392-f003:**
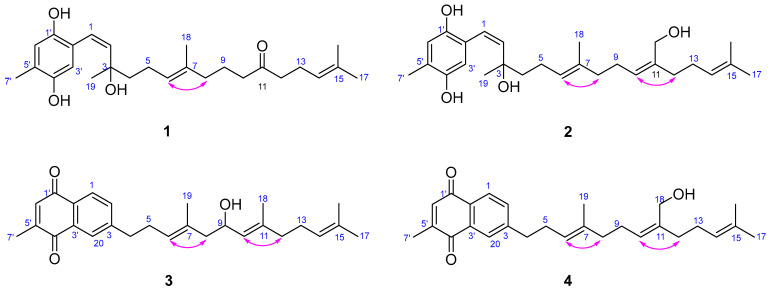
Key 1D NOEs of compounds **1**–**4**.

**Table 1 marinedrugs-22-00392-t001:** ^1^H NMR and ^13^CNMR data for compounds **1** to **4** at 600 MHz in CDCl_3_.

Position	1		2		3		4	
	δ_H_ Mult (*J*, Hz)	δ_C_, Type	δ_H_ Mult (*J*, Hz)	δ_C_, Type	δ_H_ Mult (*J*, Hz)	δ_C_, Type	δ_H_ Mult (*J*, Hz)	δ_C_, Type
1′		146.8, C		146.8, C		185.1, C		185.2, C
2′		119.7, C		119.7, C		130.5, C		130.4, C
3′	6.42, s	112.6, CH	6.42, s	112.6, CH		132.3, C		132.0, C
4′		147.6, C		147.6, C		186.1, C		186.1, C
5′		124.7, C		124. 7, C		148.1, C		148.1, C
6′	6.55, s	118.3, CH	6.55, s	118.3, CH	6.81, br.s	135.9, CH	6.81, q (1.6)	135.9, CH
7′	2.18, s	16.1, CH_3_	2.17, s	16.1, CH_3_	2.18, br.s	16.4, CH_3_	2.18, d (1,6)	16.6, CH_3_
1	6.25, d (9.8)	122.6, CH	6.25, d (9.8)	122.6, CH	7.96, d (7.9)	126.5, CH	7.96, d (7.9)	126.4, CH
2	5.52, d (9.8)	129.9, CH	5.52, d (9.8)	129.9, CH	7.52, dd (7.9, 1.8)	134.0, CH	7.52, dd (7.9, 1.8)	134.1, CH
3		78.1, C		78.1, C		148.8, C		149.1, C
4	1.63, m; 1.69, m	41.1, CH_2_	1.68, m; 1.62, m	41.0, CH_2_	2.80, t (7.6)	36.1, CH_2_	2.77, t (7.6)	36.3, CH_2_
5	2.10, m	22.8, CH_2_	2.10, m	22.8, CH_2_	2.40, q (7.4)	29.5, CH_2_	2.35, q (7.5)	29.4, CH_2_
6	5.09, tq (7.2, 1.4)	125.0, CH	5.10, tq (5.7, 1.4)	124.86, CH	5.26, m	126.6, CH	5.15, td (7.2,1.4)	123.2, CH
7		134.7, C		134.9, C		133.6, C		136.5, C
8	1.93, t (7.2)	39.1, CH_2_	1.98, t (7.5)	40.0, CH_2_	2.13, m	48.2, CH_2_	1.99, m	40.0, CH_2_
9	1.64, m	21.9, CH_2_	2.14, m; 2.10, m	26.3, CH_2_	4.42, m	66.1, CH	2.14, m	26.4, CH_2_
10	2.32, t (7.4)	42.3, CH_2_	5.28, t (7.3)	128.8, CH	5.15, d (8.2)	127.3, CH	5.27, t (7.3)	128.6 CH
11		211.4, C		138.4, C		138.5, C		138.6, C
12	2.40, t (7.4)	43.0, CH_2_	2.11, m	35.3, CH_2_	2.00, t (7.7)	39.7, CH_2_	2.11, m	35.4, CH_2_
13	2.23, q (7.4)	22. 7, CH_2_	2.10, m	27.2, CH_2_	2.07, m	26.5, CH_2_	2.11, m	27.2, CH_2_
14	5.05, tp (7.2, 1.4)	123.0, CH	5.10, tq (5.7, 1.4)	124.3, CH	5.07, tq (6.9, 1.4)	124.1, CH	5.10, tq (5.4, 1.2)	124.3, CH
15		132. 8, C		131.9, C		131.8, C		131.2, C
16	1.60, s	17.8, CH_3_	1.68, s	25.9, CH_3_	1.59, s	17.8, CH_3_	1.60, s	17.9, CH_3_
17	1.66, s	25.8, CH_3_	1.60, s	17.9, CH_3_	1.67, s	25.8, CH_3_	1.68, s	25.9, CH_3_
18	1.55, s	15.8, CH_3_	4.10, s	60.5, CH_2_	1.66, s	16.6, CH_3_	4.11, s	60.5, CH_2_
19	1.35, s	26.2, CH_3_	1.57, s	16.1, CH_3_	1.59, s	16.7, CH_3_	1.52, s	16.2, CH_3_
20			1.35, s	26.2, CH_3_	7.91, d (1.8)	126.5, CH	7.90, d (1.8)	126.6, CH

**Table 2 marinedrugs-22-00392-t002:** Inhibitory effects of **1**–**6** on six kinds of pathogenic bacteria.

MIC (µg/mL)						
Compounds	*S. Aureus*	*B. Subtilis*	*V. Harveyi*	*V. Parahaemolyticus*	*E. coli*	*P. Aeruginosa*
1	32	32	>64	32	64	16
2	>64	8	16	16	16	16
3	>64	>64	>64	>64	>64	>64
4	>64	>64	>64	>64	>64	>64
5	>64	32	>64	64	32	32
6	>64	>64	>64	>64	>64	>64
Penicillin ^a^	<0.5	<0.5	<0.5	<0.5	<0.5	<0.5

^a^ Positive control.

## Data Availability

The data presented in this study are available in the Appendix A file associated with this article.

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
