# Peer review of "New Meroterpenes from South China Sea Soft Coral Litophyton brassicum"

_marinedrugs, 2024, doi:10.3390/md22090392_

Round 1

Reviewer 1 Report

Comments and Suggestions for Authors

Review Report for marine drugs-3161260

Xiaoyi et al. conducted a chemical investigation of the soft coral Litophyton brassicum from the South China Sea. Four new meroterpenes and two were known meroterpenes were identified: brassihydroxybenzoquinone A–B,  brassinaphthoquinone A–B, and chabrolohydroxybenzoquinone G and chabrolohydroxybenzoquinone B. These compounds were comprehensively elucidated using HRMS and NMR, with UV and optical rotation measurements also included. In addition, their inhibitory activity towards tumor cells and bacteria was evaluated, with compounds 1 and 2 exhibiting moderate inhibitory activity, with MICs ranging from 8 to 64 μg/mL.

I believe this manuscript fits well within the readership of Marine Drugs. Here are a few issues that need to be addressed prior to acceptance for publication:

1.       The configuration of C-3 in both compound 1 and 2 should be determined. Mosher method may be a useful method to address this issue.

2.        The IR data for new compound 1-4 should be included.

3.       The arrows used in Figure 3 are incorrect.

The double-headed arrow used for 1-4 is inappropriate. I assume that this arrow was intended to indicate the presence of H-C correlations from both orientations. Including only the key HMBC signals is sufficient for the structure elucidation.

The dashed arrow for NOE should be solid arrow.

4.       In Table 2, the dash is used to indicate not tested or > 64

Author Response

1. The configuration of C-3 in both compound 1 and 2 should be determined. Mosher method may be a useful method to address this issue.

Response: We tried to determine the absolute configuration of C-9 in compound 3 with Mosher method, but failed. Since the C-3 of compounds 1 and 2 is on the side chain, the absolute configuration of C-3 is difficult to determine.

2. The IR data for new compound 1-4 should be included.

Response: We IR data for new compounds 1-4 was added in “Extraction and isolation” section as suggested.

3. The arrows used in Figure 3 are incorrect. The double-headed arrow used for 1-4 is inappropriate. I assume that this arrow was intended to indicate the presence of H-C correlations from both orientations. Including only the key HMBC signals is sufficient for the structure elucidation. The dashed arrow for NOE should be solid arrow.

Response: We deleted some arrows of the HMBC correlations as suggested. The arrows that have two heads were all corrected. The dashed arrow for NOE was corrected as solid arrow.

4. In Table 2, the dash is used to indicate“not tested” or “> 64”? 

Response: In Table 2, the dash was used to indicate“> 64”, We corrected it as “> 64.

Reviewer 2 Report

Comments and Suggestions for Authors

Searching new types of antibiotic is a critical endeavor for public health. In this manuscript, researchers found six meroterpenes form the Xisha soft coral Litophyton brassicum, four of which were new. Interestingly, two compounds exhibited moderate inhibitory activity against six pathogenic bacterial strains. These findings are important, and this work is suggested to be published in the forthcoming issue of this journal.

However, a few revisions were required as followings:

1. In the 1H NMR spectrum of compound 1, the characteristic methyl peaks were easily observed. It is better to provide the description of these signals in the manuscript.

2. Please remove the 1H–1H COSY correlations away from the Figure 3.

3. There were two differences between compounds 4 and 3. One was the absence of hydroxyl group at C-9, the other was the presence of hydroxyl group at C-18.

4. Have you made efforts to determine the absolute configurations of compounds 13?

5. Please provide the tR values for the compounds finally purified by HPLC.

6. Please update the information of ref. [14] as ‘…protease inhibitors. Nat. Prod. Res. 2022, 36, 2893-2896. doi: 10.1080/14786419.2021.1925892.’.

Others:

1. P1L18P7L214: brassihydroxybenzoquinone A–B (12) and brassinaphthoquinone A–B (34)brassihydroxybenzoquinones A and B (1 and 2) and brassinaphthoquinones A and B (3 and 4)

2. P2L60: repeated silica gel and reversed-phase silica gelrepeated silica gel and reversed-phase silica gel column chromatography

3. P2L62: two known compounds (56)two known compounds (5 and 6)

4. P3L77: Figure 1Figure 2’

5. P3L82: ‘The smaller coupling constants’ The smaller coupling constant’

the 1D NOE correlationsthe 1D NOE correlation

6. P4L95&P5L104: unsaturation degreesdegrees of unsaturation

7. P5L105: From the 1H NMR spectrum of 1From the 1H NMR spectrum of 2

8. P5L117: The key 1D NOE spectrumThe key 1D NOE correlations’

9. Table 2 caption: five kindssix kinds’

Comments on the Quality of English Language

There were a few typo or grammar errors, some of which were given in the comments.

Author Response

1. In the 1H NMR spectrum of compound 1, the characteristic methyl peaks were easily observed. It is better to provide the description of these signals in the manuscript.

Response:  The characteristic of methyl peaks of compound 1 in the 1H NMR spectrum was added as suggested.

2. Please remove the 1H–1H COSY correlations away from the Figure 3.

Response:  The 1H–1H COSY correlations were removed away in Figure 3.

3. There were two differences between compounds 4and 3. One was the absence of hydroxyl group at C-9, the other was the presence of hydroxyl group at C-18.

Response:  We revised the description as “The 1H and 13C NMR data (Table 1) of 4 were extremely similar to those of 3, except that the absence of hydroxyl group at C-9 in 4, and the oxygenated sp3 carbon [δC 60.5 (C-18)/δH 4.11] was a methylene in 4 rather than a methine in 3. It was deduced by analysis of the 1H-1H COSY correlations of H2-8/H2-9/H-10, and the HMBC correlations from H2-18 to C-10, C-11 and C-12.”

4. Have you made efforts to determine the absolute configurations of compounds 13?

Response: We tried to determine the absolute configuration of C-9 in compound 3 with Mosher method, but failed. Since the C-3 of compounds 1 and 2 is on the side chain, the absolute configuration of C-3 is difficult to determine.

5. Please provide the tRvalues for the compounds finally purified by HPLC.

Response: The tR values for the compounds finally purified by HPLC were provided in the “Extraction and isolation” section.

6. Please update the information of [14] as ‘…protease inhibitors. Nat. Prod. Res. 2022, 36, 2893-2896. doi: 10.1080/14786419.2021.1925892.’.

Response: Ref. [14] was revised as suggested.

7. Others:

P1L18P7L214: ‘brassihydroxybenzoquinone A–B (12) and brassinaphthoquinone A–B (34)’ → ‘brassihydroxybenzoquinones A and B (1and 2) and brassinaphthoquinones A and B (3 and 4)’

P2L60: ‘repeated silica gel and reversed-phase silica gel’ → ‘repeated silica gel and reversed-phase silica gel column chromatography’

P2L62: ‘two known compounds (56)’ → ‘two known compounds (5and 6)’

P3L77: ‘Figure 1’ → ‘Figure 2’

P3L82: ‘The smaller coupling constants’ → ‘The smaller coupling constant’

‘the 1D NOE correlations’ → ‘the 1D NOE correlation’

  1. P4L95&P4L104: ‘unsaturation degrees’ → ‘degrees of unsaturation’
  2. P5L105: ‘From the 1H NMR spectrum of 1’ → ‘From the 1H NMR spectrum of 3
  3. P5L117: ‘The key 1D NOE spectrum’ → ‘The key 1D NOE correlations’
  4. Table 2 caption: ‘five kinds’ → ‘six kinds’

Response: These errors were corrected.

Round 2

Reviewer 1 Report

Comments and Suggestions for Authors

Review Report for marine drugs-3161260 (revised version)

The authors have addressed most of my concerns, and the manuscript has been significantly improved. However, the absolute configuration issues of C-3 in both compound 1 and 2 should be determined before it can be accepted.

Comments on the Quality of English Language

None

Author Response

Dear reviewer,

We sincerely appreciate your valuable feedback on our manuscript. We attempted to utilize the TDDFT-ECD calculation method to establish the absolute configuration of the C-3 position in compounds 1 and 2. Unfortunately, the experimental ECD spectrum did not exhibit a clear Cotton effect, which is likely due to the C-3 position being located on the side chain, leading to its flexibility. As far as we know, determining the absolute configuration of C-3 remains a challenging task.